# Generation of a Perfusable 3D Lung Cancer Model by Digital Light Processing

**DOI:** 10.3390/ijms24076071

**Published:** 2023-03-23

**Authors:** Yikun Mei, Dongwei Wu, Johanna Berg, Beatrice Tolksdorf, Viola Roehrs, Anke Kurreck, Thomas Hiller, Jens Kurreck

**Affiliations:** 1Department of Applied Biochemistry, Institute of Biotechnology, Technische Universität Berlin, TIB 4/3-2, Gustav-Meyer-Allee 25, 13355 Berlin, Germany; 2BioNukleo GmbH, Ackerstr. 76, 13355 Berlin, Germany; 3PRAMOMOLECULAR GmbH, Robert-Rössle-Strasse 10, 13125 Berlin, Germany

**Keywords:** bioprinting, cancer model, digital light processing, drug testing, H358 cells, gemcitabine, apoptosis

## Abstract

Lung cancer still has one of the highest morbidity and mortality rates among all types of cancer. Its incidence continues to increase, especially in developing countries. Although the medical field has witnessed the development of targeted therapies, new treatment options need to be developed urgently. For the discovery of new drugs, human cancer models are required to study drug efficiency in a relevant setting. Here, we report the generation of a non-small cell lung cancer model with a perfusion system. The bioprinted model was produced by digital light processing (DLP). This technique has the advantage of including simulated human blood vessels, and its simple assembly and maintenance allow for easy testing of drug candidates. In a proof-of-concept study, we applied gemcitabine and determined the IC_50_ values in the 3D models and 2D monolayer cultures and compared the response of the model under static and dynamic cultivation by perfusion. As the drug must penetrate the hydrogel to reach the cells, the IC_50_ value was three orders of magnitude higher for bioprinted constructs than for 2D cell cultures. Compared to static cultivation, the viability of cells in the bioprinted 3D model was significantly increased by approximately 60% in the perfusion system. Dynamic cultivation also enhanced the cytotoxicity of the tested drug, and the drug-mediated apoptosis was increased with a fourfold higher fraction of cells with a signal for the apoptosis marker caspase-3 and a sixfold higher fraction of cells positive for PARP-1. Altogether, this easily reproducible cancer model can be used for initial testing of the cytotoxicity of new anticancer substances. For subsequent in-depth characterization of candidate drugs, further improvements will be necessary, such as the generation of a multi-cell type lung cancer model and the lining of vascular structures with endothelial cells.

## 1. Introduction

Lung cancer is among the leading causes of cancer-related death in the world, and the numbers of lung cancer cases and related mortality rates are on the rise globally. The most common type of lung cancer is non-small cell lung cancer (NSCLC, prevalence approximately 80%) [1]. Over the past two decades, the emergence of novel targeted therapies for lung cancer has led to an increasing number of tools for the clinical treatment of NSCLC [1,2]. Although up to 69% of patients with advanced NSCLC may have actionable driver mutations, many patients have little opportunity to use more effective drugs than chemotherapy [3]. One of the more recently developed treatment options is immunotherapy, which has been shown to have long-term clinical benefits [4,5]. However, despite promising initial clinical outcomes in many cases, patients may eventually develop resistance to immunotherapy. The mechanisms resulting in resistance to immunotherapy are not fully understood but include lack of tumor immunogenicity, deficiency in antigen presentation, and aberrations in signaling pathways, as well as tumor extrinsic mechanisms involved in T cell activities [6]. Some patients not only develop resistance but even experience hyperprogression [7,8]. The need for novel drugs with better efficacy that allow prolonged survival and prevent or overcome drug resistance has spurred the identification of potential targets and the development of promising therapeutic options [9,10]. Therefore, there is an urgent need to find more ways to screen and test new drugs.

Current research strategies include 2D cell cultures and animal models. However, conventional 2D cultured cancer cells cannot mimic the complexity and heterogeneity of in vivo tumors, which invariably grow in 3D conformations [11,12]. In animal models, human tumors are embedded in a microenvironment composed of animal cells [13]. This chimeric constellation is unlikely to represent the physiology of a human patient. The failure rate for the translation of drug candidates from animal testing to human treatments remains over 92% [14], and for anticancer drug candidates, even up to 97% of all substances that underwent successful pre-clinical development fail during clinical testing [15]. Although the reasons for terminating clinical trials are manifold, the high failure rate illustrates the need for better models for the pre-clinical characterization of potential anticancer drugs.

A particularly promising strategy to produce models with higher relevance for human (patho-)physiology is the generation of organ models by 3D bioprinting, which allows the arrangement of different types of (human) cells of an organ with high spatial resolution [16,17,18]. Cancer biology is one of the research fields in which bioprinting holds the most promise [19,20]. Due to its high flexibility, the technology allows the adjustment of the stiffness of the extracellular environment, the combination of various cell types, and the design of desired 3D arrangements. In a recent lung cancer model, a multicomponent bioink composed of alginate, diethylaminoethyl cellulose, gelatin, and collagen peptide was used to generate a 3D bioprinted construct, and the model was shown to be suitable for initial drug screening [21]. Virtually all of the bioprinted cancer models are composed of human cancer cells. In case non-cancerous cells are included in the model, they are usually of human origin as well, thereby overcoming the drawback of animal models, in which the human tumor is introduced into the microenvironment of mice or other animals.

Several methodologies developed for additive manufacturing have been adapted for the bioprinting of tissue models for research purposes, including microextrusion bioprinting, ink-jetting, and stereolithography [18]. Digital light processing (DLP) is a variation of stereolithography in which a complete layer is solidified at once using a digital micro-mirror device chip [22]. The method thus allows for the production of sophisticated structures at high resolution in a rapid process.

Vascularization is an essential element in artificial organ models to recapitulate natural tissue physiology. Blood vessels serve to replenish oxygen and nutrients, as well as remove waste, which is necessary for tissues and organs to maintain their function [23,24]. Cells located outside the natural diffusion range of 200–300 μm experience hypoxia and become necrotic. Hence, an appropriate perfusion system must be available to overcome this limitation and to accomplish nutrient supply in bioprinted models [25,26,27]. The aim of the present study was therefore to fabricate an easily adoptable lung cancer model with a central vascular channel. The construct was produced with a DLP bioprinter using a bioink composed of methacrylated gelatin (GelMA) and the NSCLC cell line H358. This cell line harbors a *KRAS* G12C mutation, which is an attractive target for cancer therapeutics, including MEK inhibitors [28]. The model was designed to have an inlet and an outlet to create a perfusable bioreactor system for the supply of nutrients and oxygen through a peristaltic pump. The 3D NSCLC culture system was used to investigate the efficiency of the cytostatic drug gemcitabine and characterize its mode of action. We found the perfused model to have substantially different properties than the static 3D model or a 2D monolayer culture.

## 2. Results

### 2.1. Gemcitabine Exerts Pronounced Cytotoxic Effects on H358 2D Cell Culture

For the setup of a perfused cancer model, we chose the H358 cell line, which originated from an NSCLC. As a cytotoxic drug, gemcitabine, a cytidine analog, is used for the chemotherapeutic treatment of various types of cancer. In initial experiments, 2D cultures of H358 cells were treated with different concentrations of gemcitabine. Cytotoxicity assays showed that most of the cells were viable in the absence of gemcitabine (green fluorescence), and a concentration-dependent increase in the proportion of dead cells (red fluorescence) was observed (Figure 1a). Dead cells made up a large fraction of the total number at the highest concentration used (64 nM), as indicated by the strong red fluorescence.

As gemcitabine acts as a cytotoxic agent, the impact of cell proliferation was studied by staining active DNA synthesis in EdU (5-ethynyl-2′-deoxyuridine) assays. Figure 1b shows that cells actively proliferated in the absence of gemcitabine and at low concentrations of the drug; however, at gemcitabine concentrations above 4 nM, proliferating cells were no longer observed. Previous studies have shown that the main mode of action of gemcitabine is the induction of apoptosis in cancer cells [29]. Cleaved caspase-3 is a commonly used marker to detect apoptotic cells. As can be seen in Figure 1c, the number of cells positive for cleaved caspase-3 increased with rising concentrations of gemcitabine. These initial experiments demonstrate that gemcitabine kills H358 cells and inhibits proliferation by inducing apoptosis.

### 2.2. Cell Survival after 3D Printing

In order to develop a physiologically more relevant 3D cancer model, H358 cells were mixed with GelMA and printed into 0.75 mm thick layers using the Lumen X DLP printer. The printer crosslinks GelMA by irradiation with 405 nm wavelength light. Cytotoxicity assays showed that the majority of cells survived the printing procedure (green fluorescence), and only very few dead cells (red fluorescence) were observed after 24 h (Figure 2a). Cell viability assays demonstrated that the cells’ metabolic activity remained steady for two weeks at lower cell numbers (10,000 cells) and decreased only slightly at higher cell numbers (Figure 2b). Furthermore, cells proliferated in the bioprinted 3D model, as was shown by EdU assays (Figure 2c). Taken together, these results confirmed that H358 cells survive the DLP printing process and proliferate in the 3D model.

### 2.3. Treatment of Bioprinted 3D Constructs with Gemcitabine

After having confirmed the survival of H358 cells after the printing process and during extended 3D cultivation, drug treatment experiments were also carried out in the bioprinted constructs. To this end, 0.75 mm thick structures with a total volume of 15 μL were printed by DLP. As the drug needs to penetrate the hydrogel in the 3D systems, higher concentrations are needed to induce cytotoxicity compared to monolayer cultures [30]. As can be seen in Figure 3a, cells were viable in the printed constructs in the absence of gemcitabine or at the lowest concentration of 2 µM of the applied drug. Starting at a concentration of 4 µM and continuing through 16 µM gemcitabine, an increasing number of dead, red fluorescent cells were observed. 

In line with the concentration-dependent increase of cytotoxicity, cell proliferation was inhibited as the concentration of the drug went up (Figure 3b). In parallel, cells that were positive for cleaved caspase-3 increased in a concentration-dependent manner in immunofluorescence assays (Figure 3c). Taken together, these experiments confirm that the 3D bioprinted models are suitable for studying the effects of drug treatment on H358 cancer cells and that comparable results were obtained for 2D monolayer and 3D culture, albeit substantially higher concentrations (three orders of magnitude) of gemcitabine were needed in the latter case.

### 2.4. IC_50_ of Gemcitabine for H358 Cells in 2D and 3D Culture

In order to characterize the effects of gemcitabine on H358 cells more quantitatively, IC_50_ values were determined for 2D and 3D cultures. Quantitative evaluation of cell viability as a function of drug concentration was carried out by XTT assays (Figure 4). For the 2D culture, an IC_50_ of 2.0 ± 0.7 nM was calculated. In 3D culture, gemcitabine did not kill all cells, even at the highest concentrations used. It should be noted that gemcitabine, a commonly given clinical chemotherapy drug, is generally used in combination with other antitumor drugs, as it cannot completely destroy the tumor when used alone [31,32]. The IC_50_ value can therefore only be estimated to be in the low micromolar range for the 3D construct, i.e., three orders of magnitude higher than in 2D culture. The quantitative characterization of the drug-induced cytotoxicity thus confirmed the qualitative fluorescent cytotoxicity assays described above.

### 2.5. Establishment of a Perfused Model

After having demonstrated that the DLP printing process is not overly detrimental to cell survival, we designed a model for dynamic cultivation with a perfusion system. The size of the model was minimized to save expensive bioinks and to ensure an even distribution of reagents throughout the model. The construct was printed with an oval shape, and the size of the main body of the model was 12 mm × 6 mm × 3.5 mm. In addition, an inlet and an outlet were placed on top for easy connection to the plastic tubes of the perfusion system. A curved channel with a diameter of 1.5 mm traversed the model (Figure 5a). The standard triangulation language (STL) file of the model is added as Appendix A.

The bioprinted model was connected to a peristaltic pump. Figure 5b and the Appendix A show the entire assembled perfusion setup. The distribution of media in the bioprinted model was followed by using a colored liquid (Figure 5c). As the model was originally yellow, it turned green when perfused with blue ink. The whole model was completely colored in dark green after 21 h. This confirms that the medium can nourish the cells throughout the whole construct by perfusion.

Continuous perfusion of the models washed out the yellow color of the GelMA bioink (Figure 5d). The models were then treated with an XTT reagent. As can be seen in the models on the right, perfused models were stained more intensively than models that were cultured under static conditions, indicating higher cell viability. This finding was confirmed by quantitative evaluation. The metabolic activity, as determined by the XTT assay, was increased significantly by 58.4% for models cultured under dynamic conditions compared to those that were not perfused (Figure 5e). These results demonstrate the beneficial effects of perfusion on cell viability in bioprinted 3D constructs.

### 2.6. Application of Gemcitabine by the Perfusion System

In the final step, gemcitabine was applied to the lung cancer model by the perfusion system. To determine the effect of dynamic cultivation, perfused models and models cultured under static conditions were compared. The latter were submerged in a medium containing 10 µM gemcitabine, whereas the perfused models were submerged in a medium without the drug, and gemcitabine was only applied to the model with the media stream.

Drug-induced apoptosis in H358 cells was investigated by immunohistochemistry. After four days of drug treatment, the models were fixed and embedded and then sliced by a cryotome. Figure 6a,b show that cleaved caspase-3 and cleaved PARP-1 were not detected in the absence of gemcitabine, whereas fluorescent signals appeared after drug treatment. The signals were stronger in the perfused models compared to those that were cultured under static conditions. Fluorescence microscopy images were quantitatively evaluated with the ImageJ program. Perfused Models had an approximately four times higher level of the cleaved caspase-3 signal than those cultured under static conditions (Figure 6c). Likewise, the signal of cleaved PARP-1 was approximately six times higher for the dynamic cultivation (Figure 6d).

## 3. Discussion

Three-dimensional organ models have developed as an alternative approach to conventional animal experiments, which are composed of human cells and, at the same time, may help to reduce the number of animal experiments [33]. Bioprinting is one of the most promising technologies for producing disease models for oncology research and can be used to develop anticancer substances and optimize irradiation procedures [20,34]. Compared to other techniques for the generation of 3D cultures, such as organoid formation, bioprinting has the advantage of allowing for the computer-based design of 3D objects with a predefined structure [18,35,36]. This unique feature can be used to include vascular channel systems in the models, which enhance the physiological relevance of the models. Different strategies have been developed to this end. In many cases, fugitive inks were used that produced a distinct vascular pattern during the printing process and were removed post-printing during the culture period [37,38]. Compared to widely used extrusion-based bioprinting, DLP combines the advantages of extraordinarily high resolution and high stability of the produced constructs. Fugitive inks can still be used to create tubs [39], but we found the stiffness of the constructs to be sufficient to generate a vascular structure without the need for additional materials.

While we and others have presented bioprinted models composed of lung cancer cells by extrusion-based technologies before [21,40,41], the present study is, to the best of our knowledge, the first lung cancer model produced by DLP bioprinting. The particularly high resolution of this methodology allowed the inclusion of the predesigned vascular structure. The model was then connected to a perfusion system to study the mode of action of a cytostatic drug. The commercially available Lumen X DLP printer (Cellink, Gothenburg, Sweden) can produce very complex structures with high resolution. As mentioned before, crosslinking of methacrylated gelatin generates very stable structures so that a channel can be included in the construct. In our model, we implemented a single channel that winds through the construct to nourish all parts of the model. The model has a high toughness, can be bent, and is not easily damaged. In human tumors, secretion of extracellular matrix proteins and crosslinking of the fibrillar collagen matrix also usually lead to a comparatively high tissue stiffness [42].

The model was designed with an inlet and an outlet to which plastic tubes can easily be connected. Circulating media with a peristaltic pump allowed comparison of 3D models cultured under static and dynamic conditions, respectively, clearly demonstrating that perfusion increases cell viability during long-term cultivation. In a previous study, we found the addition of further cell types to a bioprinted model increased cell viability over the course of four weeks, even in a static culture [41]. Thus, the combination of multiple cell types in the vascularization approach described here can be expected to further extend the period of cultivation. Furthermore, increasing cell density would increase the physiological relevance of the model, as the current density is still quite low compared to primary tissues. The resolution of DLP may, however, be lowered by the increase in light scattering at higher cell densities. In a recent study, You et al. reported the addition of iodixanol to reduce scattering and improve the quality and resolution of biofabrication procedures based on DLP [43].

In our experiments, the anticancer drug gemcitabine was applied through the perfusion system. The control models were cultured under static conditions by immersion in a medium containing the drug. In contrast, the perfused models received the active substance only through the media flow. This reflects the administration of a drug through the bloodstream in a human patient.

A challenge for all types of microphysiological systems is the emergence of bubbles, which can affect cell viability and efficacy of applied substances, and may, over the course of long-term cultivation, block circulation partially or entirely. Avoidance and removal of bubbles are challenging tasks [44,45]. In our perfusion system, we did not observe the appearance of air bubbles. The force of the peristaltic pump was uniform and stable. Upon connection of the bioprinted organ model to the media circuit, an initially high flow rate was used to fill the entire tube quickly and to eliminate bubbles. Furthermore, with the help of an external syringe that was connected to a side port, the tubing from the peristaltic pump to the incubator can be filled with solution (culture medium, PBS, drug solution, etc.). Following the initial flushing of the system at a high flow rate of 600 µL/min, the flow rate was halved to prevent damage to the system. Over a time period of four days, perfusion was smooth, and no stagnation or bubble formation was observed. Upon longer cultivation, evaporation resulted in the reduction of the volume of the fluid, and the media was replenished by the external syringe to reduce the risk of contamination.

For proof-of-principle experiments, gemcitabine was used to test the reaction of the perfused organ model to treatment with a cytostatic agent. Many chemotherapeutic drugs achieve their effect by inducing tumor cell apoptosis [46]. Caspase-3 is a key molecule in the signaling pathway of apoptosis in cancer cells. The inactive pro-caspase is activated by the proteolytic generation of cleaved caspase-3. The active form then acts as a proteolytic enzyme itself and activates further downstream factors, including PARP-1, which is involved in DNA damage and repair [47,48]. Immunohistochemical analysis of these two markers, cleaved caspase-3 and PARP-1, was carried out to detect apoptotic processes induced by gemcitabine treatment of H358 cells. In dose-response assays, a clear concentration-dependent effect of drug treatment was observed. In 2D culture, an IC_50_ as low as 2.5 nM was determined, which is in line with IC_50_ reports by [49]. For 3D cultures, approximately 1000-fold higher concentrations were required to induce cell death. This observation can be explained by the dense network formed by the hydrogel and the cells, which reflects the situation of the dense tumor tissue in patients. This density also frequently hinders the penetration of anticancer drugs into cancer tissues. We made a similar observation in our previous study [30], and others also reported comparable results. For example, Imamura et al. found cell lines that developed dense multicellular spheroids to develop greater resistance to paclitaxel and doxorubicin compared to the 2D cultured cells [50]. Similarly, various head and neck squamous cell carcinoma cells were reported to show decreased sensitivity to cisplatin and cetuximab in 3D [51]. Prostate cancer cells DU145 and glioblastoma cells U87 were considerably more resistant to dasatinib-induced toxicity than corresponding cells cultured in monolayer, which was explained by the very limited ability of the drug to penetrate into the 3D model [52]. Thus, cells in 3D constructs are often observed to be less sensitive to drug treatment than monolayer cultures. Based on their in-depth analysis, Imamoura et al. concluded that 3D models simulate tumor characteristics in vivo with respect to hypoxia, dormancy, anti-apoptotic features, and their resulting drug resistance better than 2D cultured cells.

Even at the highest concentration of gemcitabine used in the present study, a substantial fraction of the cells survived. Although gemcitabine is a well-established anticancer drug [53,54], it is commonly used in combination with additional drugs to improve the antitumor activity. The partial cytostatic effect observed in 3D culture is thus closer to the biological situation in human patients than the high efficiency of gemcitabine found for 2D cultures.

In order to improve the physiological relevance of the model further, the channel needs to be seeded with endothelial cells, e.g., HUVECs, in further optimization experiments. Previous studies have shown that a tight endothelial lining can be achieved by post-printing seeding of cells and slow rotation of the model to ensure even distribution of the cells on the entire vessel wall [38]. The tight endothelium will hinder substances from easily penetrating into the model and thereby more realistically simulate the pharmacokinetic features of human patients. Interestingly, the addition of pro-angiogenetic factors induced pronounced microvessel formation sprouting from the designed vascular structure that was lined by endothelial cells [38].

In addition to the endothelial lining, further improvements will be necessary to better approximate a real patient’s tumor. In a recent study, a breast cancer model was generated that used patient-derived cancer organoids instead of established tumor cell lines [55], which is a major advancement on the path to obtaining clinically relevant data. The proliferation, metabolism, and invasive potential of cancer cells are largely determined by the tumor microenvironment. The composition of stromal cells, such as fibroblasts, adipocytes, endothelial cells, and pericytes, as well as the extracellular matrix, needs to be modeled to fully recapitulate the characteristics of a biological tumor [19]. Furthermore, several studies have used decellularized extracellular matrix as a natural cell environment in the bioink [56,57]. Another challenge that is relevant to all types of in vitro 3D models is the inclusion of an immune system to simulate the natural immune reaction to malignant cells. Finally, the interaction between different organs is essential to mimic in vivo physiology. Microphysiological devices have been developed that connect multiple types of organoids [58]. There is great potential in combining these organ-on-a-chip systems that usually employ 2D cultures or simple spheroids/organoids with bioprinting technologies, which can arrange complex 3D organ systems with high resolution.

While such sophisticated models will be required for an in-depth understanding of tumor biology and detailed characterization of the mode of action of new drugs, simpler models can help to support initial substance screening. Our model, which uses a commercially available bioprinter and a standard bioink that can be purchased, combined with the freely available STL code of the 3D construct, can be easily adapted by other researchers to produce a vascularized cancer model. Due to the high stability of crosslinked GelMA, printing of the models is straightforward and does not require the use of fugitive inks. Likewise, the peristaltic pump and tubes are standard equipment and are easily available. 

Altogether, our study clearly demonstrates substantial differences between results obtained with cells cultured in 2D monolayers and those included in a 3D arrangement as in biological tissue. Most importantly, we have shown the advantages of connecting the model to a perfusion system to carry out experiments under dynamic conditions. For a proof-of-concept study, we have used an approved cytostatic agent and demonstrated that the model can be employed to study the efficiency and mechanism of action of proliferation inhibitory substances. The syringe port in the perfusion circuit also allows repeated dosing of the drug, which is commonly performed in clinical chemotherapeutic settings.

## 4. Materials and Methods

### 4.1. Cell Culture

The human non-small cell lung cancer cell line H358 was purchased from ATCC (ATCC-CRL-5807; ATCC, Manassas, VA, USA) and cultured in RPMI-1640 medium containing 10% fetal bovine serum (FBS; c.c.pro, Oberdorla, Germany), 2 mM L-glutamine (Biowest, Nuaillé, France), 1% non-essential amino acids (NEAA, Biowest), and 1% penicillin-streptomycin (Biowest) and maintained at 37 °C in a humidified atmosphere with 5% CO_2_. When confluent, cells were washed with phosphate-buffered saline (PBS, Biowest) and then harvested using trypsin-EDTA (Biowest).

### 4.2. Cytotoxicity Assay

In order to analyze the cell survival after printing and drug treatment, cytotoxicity tests were performed. A viability/cytotoxicity kit (Thermo Fisher Scientific, Waltham, MA, USA) was used according to the manufacturer’s instructions. Analysis of the 2D cells or 3D models using an inverted fluorescence microscope (Observer Z1, Zeiss, Jena, Germany) was conducted after a 1 h incubation in phenol red-free RPMI medium containing 2 μm calcein-AM and 2 μm ethidium homodimer-1.

### 4.3. Cell Viability Assay

Cell viability was determined by XTT assays (2,3-Bis-(2-Methoxy-4-Nitro-5-Sulfophenyl)-2H-Tetrazolium-5-Carboxanilide, Alfa Aesar, Ward Hill, MA, USA). The metabolization of the tetrazolium salts was measured at different time points. XTT reagent (1 mg/mL in RPMI, Biowest) and phenazine methyl sulfate (PMS, 3.83 mg/mL in PBS, AppliChem, Darmstadt, Germany) were prepared in a 500:1 ratio, and 100 μL were added to the 2D cell culture in each well of a 96-well plate and incubated for 4 h at 37 °C and 5% CO_2_. Absorbance was measured at 450 nm and 620 nm reference using a Sunrise microplate reader (Tecan, Männedorf, Switzerland). For 3D constructs, the XTT/PMS reagent was added in a fivefold excess to the bioink for 4 h, and the absorbance of the supernatant was measured as described above. Alcohol-treated cells were used as background control. Cell viability was calculated using the following formula:Cell Viability = Absorbance Test Wells − Absorbance Background Wells

Half-maximal inhibitory concentration (IC_50_) values were calculated from nonlinear regression curves (dose-response curves) of dose-response data using GraphPad Prism 7 (GraphPad, La Jolla, CA, USA). All experiments were performed at least three times.

### 4.4. Immunofluorescence Staining of 2D and 3D Cultures

Samples were first washed with PBS and fixed in 4% formaldehyde (Carl Roth, Karlsruhe, Germany) for 10 min (2D) or 15 min (3D) at room temperature. After washing twice with PBS, the samples were treated with 1% bovine serum albumin (BSA) blocking solution containing 0.1% (*v/v*) Triton X-100 (Carl Roth) for 1 h. Afterward, the blocking solution was discarded, and samples were incubated with the respective primary antibody (anti-cleaved PARP (Cleaved PARP (Asp214) (D64E10) XP^®^ Rabbit mAb #5625, 1:400, Cell Signaling Technology, Danvers, MA, USA; anti-cleaved caspase3, 1:400, Cell Signaling Technology) overnight at 4 °C. The samples were then washed three times with PBS and incubated with the corresponding secondary antibodies (goat anti-mouse Alexa Fluor 594, 1:500, Invitrogen, Carlsbad, CA, USA; goat anti-rabbit Alexa Fluor 488, 1: 500, Invitrogen) for 1 h at room temperature. After washing twice with PBS, nuclear staining was performed with 1 μg/mL of 40,6-diamidino-2-phenylindole (DAPI, Sigma-Aldrich, St. Louis, MO, USA) for 30 min, and samples were washed once again with PBS. Stained samples were analyzed by fluorescence microscopy (Observer Z1, Carl Zeiss).

### 4.5. Cryosectioning and Immunofluorescence Assays

After washing the models with PBS, they were placed in 12-well plates (Corning, Glendale, AZ, USA) and fixed with 4% paraformaldehyde for 30 min. Following another washing step with PBS, the constructs were incubated with infiltration solution (10% BSA) overnight at 4 °C. The constructs were then removed from the infiltration solution, rinsed once with PBS, cross-cut to size in the direction of the internal tubing, and transferred to a flat-side-down freezing mold. Each freezing mold was then filled with optimal cutting temperature compound (OCT), and the molds were placed at −80 °C to freeze overnight. After solidification, samples were transferred to −20 °C and stored until cryosectioning. Frozen blocks were routinely cryosectioned using a cryostat (Leica CM 1850, Wetzlar, Germany) at −20 °C. Sections were cut into 14 μm slices, placed on glass slides, air-dried, and stored at −20 °C until staining. For staining, the sections were washed once for 5 min in PBS. Afterward, the samples were treated with 5% BSA blocking solution containing 0.1% (*v/v*) Triton X-100 (Carl Roth) for 1 h. After removal of the blocking solution, the corresponding primary antibodies (anti-Cleaved PARP (Cleaved PARP (Asp214) (D64E10) XP^®^ Rabbit mAb #5625, 1:400, Cell Signaling Technology; anti-Cleaved Caspase3, 1:400, Cell Signaling Technology) diluted in blocking solution were added to each section and incubated overnight at 4 °C. The primary antibody solution was removed, and sections were washed once again for 5 min with a blocking solution. The secondary antibody (Goat anti-Rabbit IgG (H+L) Highly Cross-Adsorbed Secondary Antibody, Alexa Fluor™ 488, 1:500, Thermo Fisher Scientific, catalog # A-11034, RRID AB_2576217), diluted in blocking solution, was added dropwise to the sample, and incubated at room temperature for 1 h in the dark. After 5 min wash in PBS, Mount FluorCare DAPI (Art. No. HP20.1, Carl Roth) was added dropwise to each section, which was then covered with a coverslip. After 40 min, the samples were imaged by fluorescence microscopy (Observer Z1, Carl Zeiss), and positive signals for cleaved caspase-3 or cleaved PARP-1 were normalized to the total number of cells as determined by DAPI signal using the software ImageJ (1.53e, National Institutes of Health, Bethesda, MD, USA).

### 4.6. Proliferation Assay

Cell proliferation was measured in monolayer cultures as well as in 3D models using the EdU (5-ethynyl-2’-deoxyuridine) Cell Proliferation Imaging Kit (Invitrogen) according to the manufacturer’s instructions. The analog is incorporated during DNA synthesis. After 4 days of drug treatment, 20 µM of EdU were added to the 2D or 3D cultures. Samples were incubated in a humidified CO_2_ incubator at 37 °C (2 h for 2D cells, 4 h for 3D models), fixed with Click-iT fixative for 15 min at room temperature, and rinsed in PBS containing 3% BSA. Samples were permeabilized with 0.5% Triton X-100 for 30 min at room temperature, rinsed in PBS containing 3% BSA, and incubated with Click-iT reaction mix (containing Alexa Fluor 488 azide) for 30 min at room temperature in the dark. The samples were washed twice in PBS containing 3% BSA, 1x Hoechst^®^ 33342 solution was added to each well, incubated in the dark for 30 min, and then analyzed by fluorescence microscopy (Observer Z1, Carl Zeiss).

### 4.7. Bioprinting

The GelMa PhotoInk (D16110022026, Cellink, Gothenburg, Sweden) was thawed in a 37 °C water bath before printing. The Lumen X Digital Light Processing bioprinter (Cellink) was preheated with a set temperature of 65 °C. After mixing the hydrogel well with H358 cells, it was added to a disposable PDMS dish (D16110020872, Cellink). The cell concentration was 11,000 cells/μL. The following parameters were set for printing: 7.25 s exposure, 100 µm resolution, and 54% projector power level. The model was designed to contain two ports on the top for connection to plastic tubes. Inside the model, a curved 1.5 mm diameter channel was placed for perfusion with medium (Figure 5a). The STL file of the model is provided as Appendix A. After printing, the models were washed in PBS for 10 min and transferred into 12 well plates. Per well, 2 mL of medium was added, and the models were incubated overnight at 37 °C and 5% CO_2_ for subsequent experiments.

### 4.8. Perfusion System

The supportive parts, plastic tubes (Fluidflex silicone hose, Pro Liquid), and peristaltic pump (Ismatec, Idex) were sterilized with alcohol prior to the experiments and rinsed three times with PBS to remove residual alcohol. The completely assembled perfusion system is shown in Figure 5c. The well B1 was used as a reservoir, containing 2 mL of culture medium with or without 10 μm gemcitabine. The bioprinted 3D model was placed in well B2, submerged in media that did not contain gemcitabine, and connected to the plastic tubes using a supportive bridge (white) for increased stability. The reservoir outlet (B1) and the inlet of the model (B2) were connected with a short tube, while the outlet of the model and the inlet of the reservoir were connected to the peristaltic pump by longer tubes. The flow direction was set so that the 3D model was provided with media from the reservoir. The design of the inlet and outlet of the model matched the size of the plastic tubes so that they were tightly fixed without the use of additional glue. After assembling the entire perfusion system, the model was initially perfused for 2 min at a high flow rate of 600 μL/min. A 5 mL syringe containing the same culture medium as in the reservoir was connected to the side port of the peristaltic pump (Figure 5c). During the initial perfusion, 3 mL of the culture medium inside the syringe was slowly injected into the circulation, allowing the culture medium to fill the entire tubing and model and eliminate all bubbles in the reservoir. After the 2 min initiation phase, the flow rate of the peristaltic pump was then reduced to 300 μL/min. The peristaltic pump can perfuse two circuits simultaneously, so the perfused control 3D model and the treated 3D model were subjected to the same flow rate. Finally, the entire perfusion system was placed in an incubator (37 °C, humidified atmosphere, 5% CO_2_).

### 4.9. Statistics

Results are shown as the means ± standard deviation (SD) of at least three independent experiments. Statistical analysis was performed using GraphPad Prism 7 (GraphPad). One-way ANOVA was used for the analysis of variance to compare groups. Statistical significance was accepted at the levels of ** p* < 0.05, *** p* < 0.01, **** p* < 0.001, and ***** p* < 0.0001.

## 5. Conclusions

The present study describes the generation of a perfused lung cancer model by digital light processing bioprinting technology. Importantly, the model can easily be adapted by other groups working in the field of cancer research and 3D culture, as we have provided all the necessary information to reproduce the model with commercially available devices and materials. We demonstrate that cells survive the printing procedure and that the models can be used to study the toxicity of cytostatic substances. Furthermore, the inclusion of a channel in the model allows connection to a perfusion system that has some properties of the vascular system in living organisms. Compared to static cultivation, perfusion of the model resulted in higher cell viability. In a proof-of-concept study, we used this setup to study the activity of an approved cancer drug and to study its mechanism of action. We therefore consider the model to be suitable for testing new substances with potential antitumor activity. Further improvements of the model, such as endothelial lining, will be included in the future to further increase its physiological relevance.

## Figures and Tables

**Figure 1 ijms-24-06071-f001:**
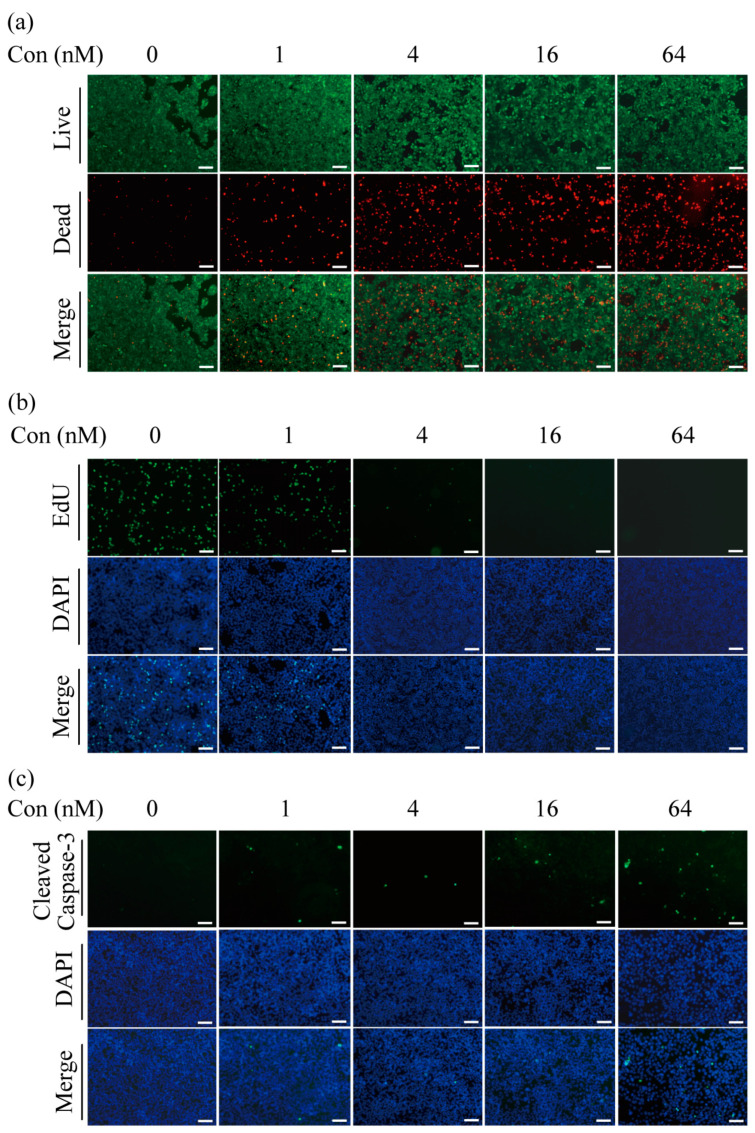
Effects of gemcitabine on H358 cells in a 2D monolayer culture. Increasing concentrations of gemcitabine were added to the culture medium as indicated. After four days of treatment, the 2D cultures were analyzed. (**a**) Cytotoxicity assays were carried out by staining cells with calcein-AM and ethidium homodimer-1. Living cells appear in green under the fluorescence microscope and dead cells in red. (**b**) The effects of gemcitabine treatment on cell proliferation were determined by EdU assays. (**c**) Immune fluorescence staining for cleaved caspase-3 was used to identify apoptotic cells following gemcitabine treatment. Representative data from three independent experiments are shown. Scale bar: 200 μm.

**Figure 2 ijms-24-06071-f002:**
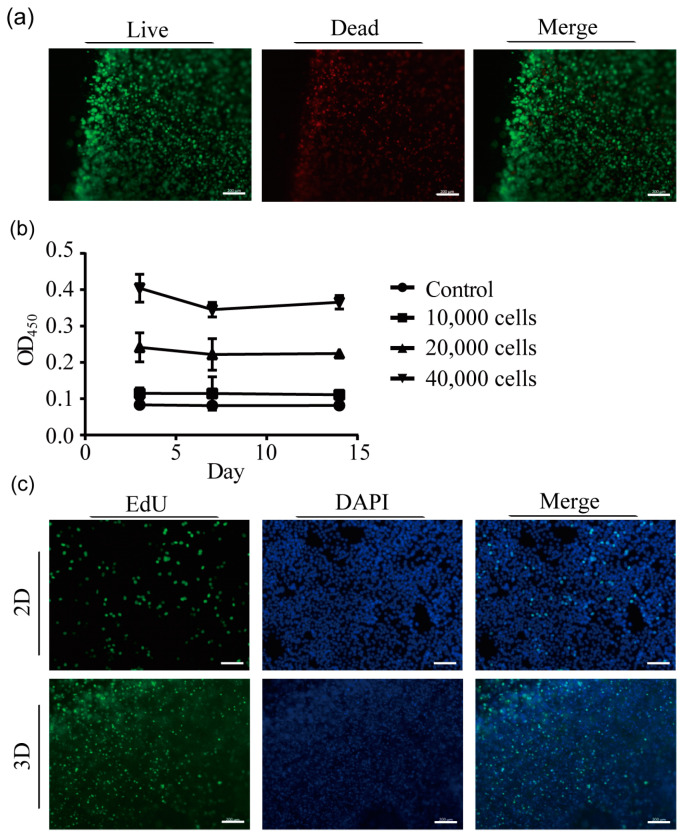
Cytotoxicity, cell viability and proliferation of H358 cells in 3D printed models. (**a**) Cytotoxicity assays were carried out by staining cells with calcein-AM and ethidium homodimer-1. Living cells appear in green under the fluorescence microscope and dead cells in red. (**b**) The metabolic activity of H358 cells in 3D models measured at three time points over a time period of two weeks was determined by cell viability (XTT) assay and plotted. As a control, the values for cells that were treated with 75% alcohol are given. Data from three independent experiments are presented as mean ± SD. (**c**) Proliferation of H358 cells in the bioprinted 3D models was determined by EdU assays 24 h after printing. Proliferating cells appear in green. Cell nuclei are blue after DAPI staining. Representative data from three independent experiments are shown. Scale bar: 200 µM.

**Figure 3 ijms-24-06071-f003:**
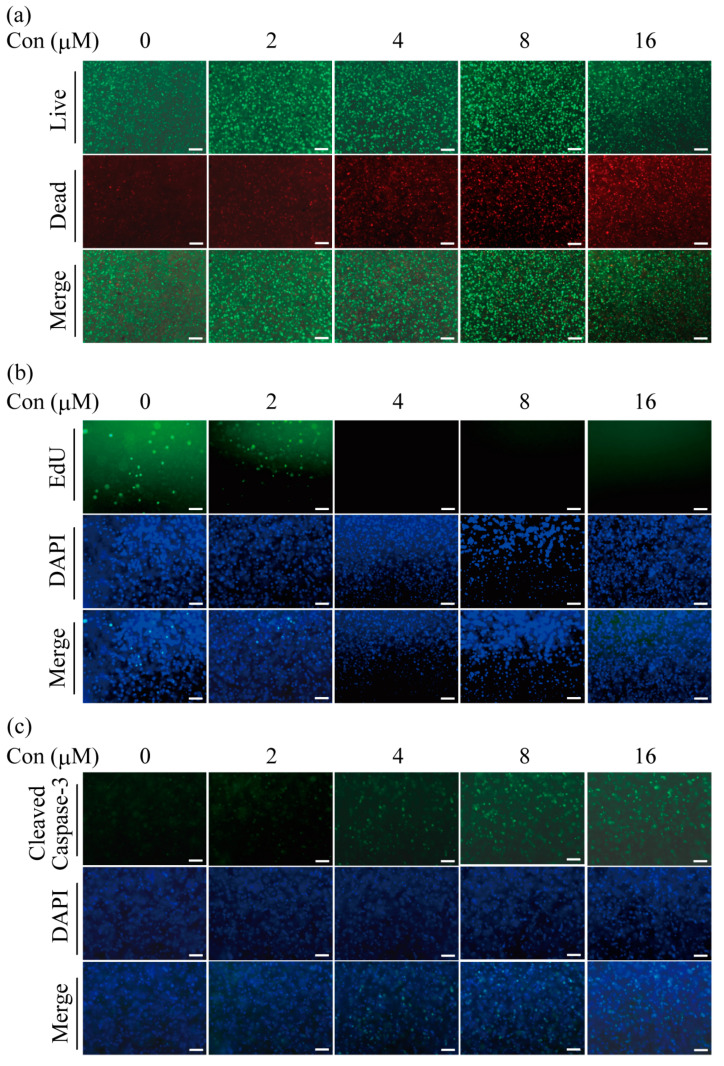
Effects of gemcitabine on H358 cells in 3D printed models. Increasing concentrations of gemcitabine were added to the culture medium as indicated. After four days of treatment, the 3D cultures were analyzed. (**a**) Cytotoxicity assays were carried out by staining cells with calcein-AM and ethidium homodimer-1. Living cells appear in green under the fluorescence microscope and dead cells in red. (**b**) Proliferation of H358 cells in the bioprinted 3D models after treatment with increasing concentrations of gemcitabine was determined by EdU assays. Proliferating cells appear in green. Cell nuclei are blue after DAPI staining. (**c**) Induction of apoptosis by gemcitabine treatment is shown, as detected by immunofluorescence staining for cleaved caspase-3. Representative data from three independent experiments are shown. Scale bar: 200 µM.

**Figure 4 ijms-24-06071-f004:**
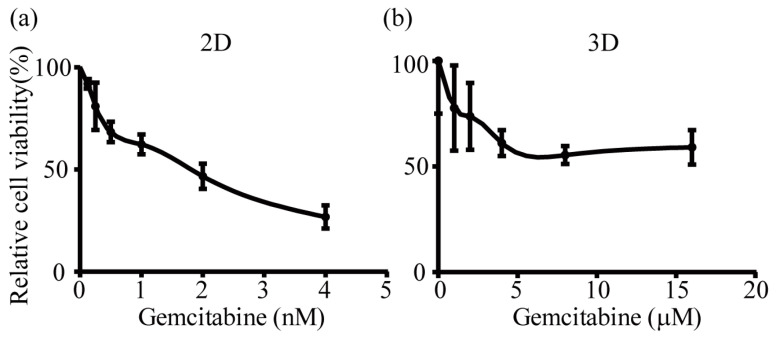
Concentration dependency of gemcitabine-induced cytotoxicity as determined by XTT assays. H358 cells in 2D (**a**) and 3D (**b**) cultures were treated with gemcitabine at nanomolar concentrations for 2D and micromolar concentrations for 3D for 24 h. Data from three independent experiments are presented as mean ± SD.

**Figure 5 ijms-24-06071-f005:**
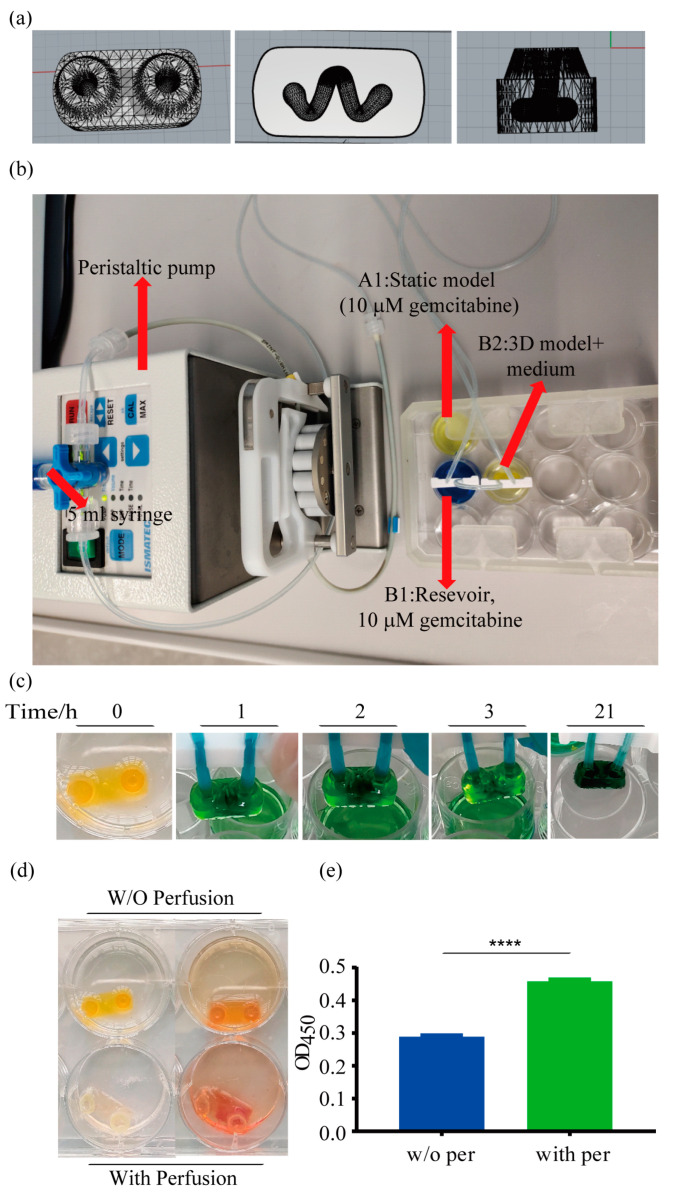
The 3D model design and perfusion system. (**a**) The 3D model was designed with the Rhino 6 program (left: top view of the model; middle: sectional view of the internal channel of the model; right: front view of the model). The Lumen X DLP bioprinter was then used to produce the 3D model using a GelMA-based bioink. (**b**) The whole perfusion system, including a peristaltic pump connected to the 3D lung cancer model. A side port allows a connection to a syringe. (**c**) The yellow models were perfused with blue ink. After 21 h of perfusion with blue ink, the dark green color indicates a thorough mix of the yellow model and the blue ink. (**d**) Models were then cultured in regular media of orange-red color. Untreated models are shown on the left, while the models on the right were incubated with XTT reagent, which becomes orange when converted by metabolically active cells. The two models on top were cultured under static conditions, while the lower models were perfused. (**e**) Viability of cells in the model after perfusion for four days was quantified by XTT assays. The blank value of the untreated models was subtracted from the models incubated with the reagent. Data are presented as mean ± SD; n = 4. ***** p* < 0.0001.

**Figure 6 ijms-24-06071-f006:**
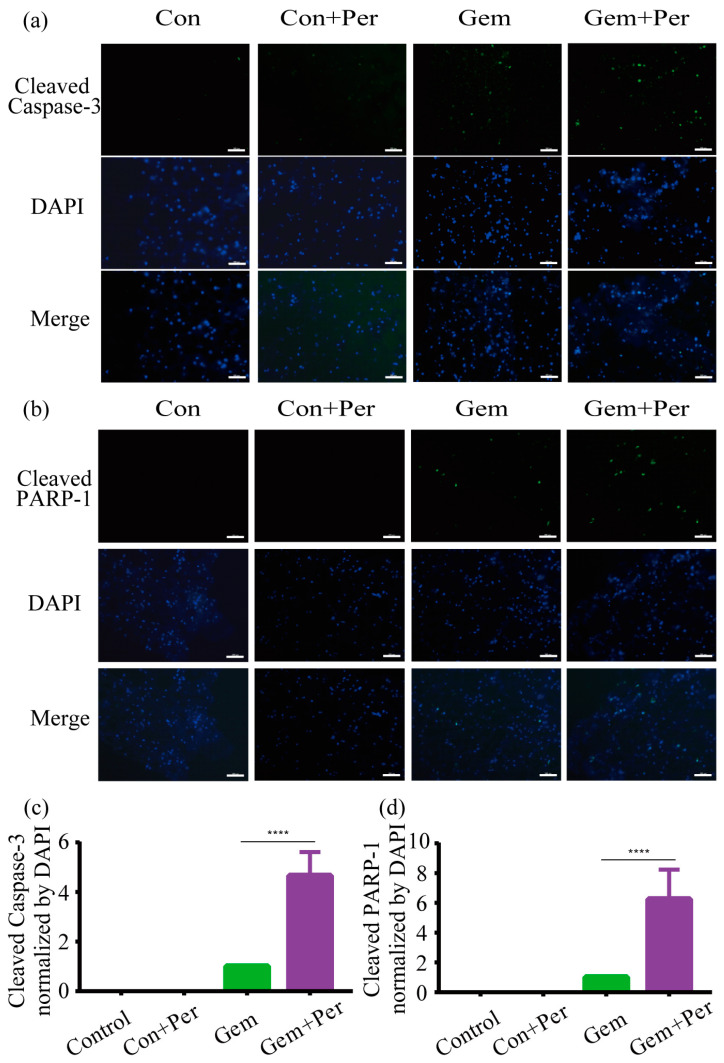
Gemcitabine-induced apoptosis in H358 cells cultured in bioprinted 3D models under static and dynamic conditions. (**a**,**b**) Immunohistochemical analysis of cleaved caspase-3 (**a**) and PARP-1 (**b**) in cryosections of the bioprinted model. Models were treated with 10 μm gemcitabine for four days and then fixed, embedded, and sliced by cryosectioning. Representative data from three independent experiments are shown. Scale bar: 200 µm. (**c**,**d**) Quantification of fluorescent cells by ImageJ 1.53e. Positive signals for cleaved caspase-3 (**c**) and PARP-1 (**d**) were normalized to the total number of cells, as determined by DAPI staining. The value for gemcitabine-treated models under static conditions was set to 1 in order to present the perfusion data relative to it. Data are presented as mean ± SD; n = 3. ***** p* < 0.0001.

## Data Availability

Not applicable.

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
