# Peer review of "Generation of a Perfusable 3D Lung Cancer Model by Digital Light Processing"

_ijms, 2023, doi:10.3390/ijms24076071_

Round 1

Reviewer 1 Report

This manuscript reports the fabrication of perusable 3D lung cancer model and evaluation of its suitability using gemcitabine. This is an interesting approach to develop dynamic 3D lung cancer model which could reduce the dependency on animal model. There are some issues that should be addressed prior to acceptance for publication. 

1) Authors should strengthen the introduction by emphasising the flexibility and advantages of bioprinting in fabricating organs. The authors need to highlight the current achievements of this bioprinting designing tumor model.

2) Why alcohol-treated cells were used as background control? It would be great if the authors could include the apoptosis assay (quantitative data) which could support the previous toxicity experiment. Is quality control measurement conducted towards the bioprinting model such as optimization of the printing process and numbers of apoptotic cells? 

3) The discussion section needs significant improvement. These include the discrepancy of IC50 data between 2D and 3D culture when treated with gemcitabine. The authors need to include the comparison of characteristics between 3D culture and the real tumor. Authors should include the limitation or challenges in bioprinting. This information is important to enhance the overall quality of the discussion.

Author Response

This manuscript reports the fabrication of perusable 3D lung cancer model and evaluation of its suitability using gemcitabine. This is an interesting approach to develop dynamic 3D lung cancer model which could reduce the dependency on animal model. There are some issues that should be addressed prior to acceptance for publication.

1) Authors should strengthen the introduction by emphasizing the flexibility and advantages of bioprinting in fabricating organs. The authors need to highlight the current achievements of this bioprinting designing tumor model.

We made substantial additions to the introduction to make the advantages and flexibility of biofabricated organ models clear and to present our current achievements. Further additions concerning the current status of the bioprinting technology were made in the discussion section, as also requested by reviewer 3. The following paragraph was added to the introduction:

A particularly promising strategy to produce models with higher relevance for human (patho-) physiology is the generation of organ models by 3D bioprinting which allows the arrangement of different types of (human) cells of an organ with high spatial resolution is a promising technology to produce organ models composed of human cells [15-17]. Cancer biology is one of the research fields in which bioprinting holds the most promise [18, 19]. Due to its high flexibility, the technology allows the ad-justment of the stiffness of the extracellular environment, the combination of various cell types and the design of desired 3D arrangements. In a recent lung cancer model, a multicomponent bioink composed of alginate, diethylaminoethyl cellulose, gelatin, and collagen peptide was used to generate a 3D bioprinted construct, and the model was shown to be suitable for initial drug screening [20]. Virtually all of the bioprinted cancer models are composed of human cancer cells. In case non-cancerous cells are in-cluded in the model, they are usually of human origin as well, thereby overcoming the drawback of animal models, in which the human tumor is introduced into a microenvironment of mice or other animals.

2) Why alcohol-treated cells were used as background control? It would be great if the authors could include the apoptosis assay (quantitative data) which could support the previous toxicity experiment. Is quality control measurement conducted towards the bioprinting model such as optimization of the printing process and numbers of apoptotic cells?

As can be seen in Fig. 5c and d, the models are colored by the photoabsorber. The colors disturb XTT measurement. It was therefore necessary to subtract the OD of the photoabsorber from the measurements to obtain metabolic activity of the cells. Alcohol-treated cells are no longer metabolically active, so that this control gives the OD of the pure photoabsorber. In our study, cell viability and apoptosis assays serve as quality control for our model. As can be seen in Figure 3, the model contains almost exclusively viable cells, and virtually no apoptotic cells were stained in the absence of the cytotoxic substance. The printing conditions used thus ensure optimal condition for the cells.

3) The discussion section needs significant improvement. These include the discrepancy of IC50 data between 2D and 3D culture when treated with gemcitabine. The authors need to include the comparison of characteristics between 3D culture and the real tumor. Authors should include the limitation or challenges in bioprinting. This information is important to enhance the overall quality of the discussion.

We included a full paragraph on the explanation of the differences between 2D and 3D culture with multiple references and compared the situation to that of real tumors: This observation can be explained by the dense network formed by the hydrogel and the cells, which reflects the situation of the dense tumor tissue in patients. This density also frequently hinders the penetration of anticancer drugs into cancer tissues. We made a similar observation in our previous study [29] and others also reported comparable results. For example, Imamura et al. found cell lines which developed dense multicellular spheroids to develop greater resistance to paclitaxel and doxorubicin com-pared to the 2D cultured cells [49]. Similarly, various head and neck squamous cell car-cinoma cells were reported to show decreased sensitivity to cisplatin and cetuximab in 3D [50]. Prostate cancer cells DU145 and glioblastoma cells U87 were considerably more resistant to dasatinib-induced toxicity than corresponding cells cultured in monolayer, which was explained by the very limited ability of the drug to penetrate into the 3D model [51]. Thus, cells in 3D constructs are often observed to be less sensitive to drug treatment than monolayer cultures. Based on their in-depth analysis, Imamoura et al. concluded that 3D models simulate tumor characteristics in vivo with respect to hypoxia, dormancy, anti-apoptotic features and their resulting drug resistance better than 2D cultured cells.

Furthermore, we included a paragraph on limitations of bioprinting and issues that should be addressed to make the model physiologically more relevant:

In addition to the endothelial lining, additional improvements will be necessary to better approximate a real patient’s tumor. In a recent study, a breast cancer model was generated that used patient-derived cancer organoids instead of established tumor cell lines [54], which is a major advancement on the path to obtaining clinically relevant data. Proliferation, metabolism and invasive potential of cancer cells are largely deter-mined by the tumor microenvironment. The composition of stromal cells, such as fibroblasts, adipocytes, endothelial cells and pericytes, as well as the extracellular ma-trix, needs to be modeled to fully recapitulate the characteristics of a biological tumor [18]. Furthermore, several studies have used decellularized extracellular matrix as a natural cell environment in the bioink [55, 56]. Another challenge that is relevant to all types of in vitro 3D models is the inclusion of an immune system to simulate the natural immune reaction to malignant cells. Finally, the interaction between different organs is essential to mimic in vivo physiology. Microphysiological devices have been developed that connect multiple types of organoids [57]. There is a great potential in combining these organ-on-a-chip systems that usually employ 2D cultures or simple spheroids/organoids with bioprinting technologies, which can arrange complex 3D organ systems with high resolution.

We are thankful for the suggestions made by reviewer 1 and think that these additions have strengthened our manuscript.

Reviewer 2 Report

The paper presented by Yikun Mei et al in the paper provides the technology for the generation of bioprinted lung cancer models by digital light processing. Authors emphasize that the advantage of their method is the possibility of simulated human blood vessels within the 3D model, however, the results do not support that conclusion. Moreover, the overall merit of the results seems not to be more proper for technical/methodological journals rather than for the IJMS aims and scope. Publication in IJMS requires sequential revision including experiments that will provide much more biological meaning, as well as clearly indicate future perspectives for its application for the study.

Author Response

The paper presented by Yikun Mei et al in the paper provides the technology for the generation of bioprinted lung cancer models by digital light processing. Authors emphasize that the advantage of their method is the possibility of simulated human blood vessels within the 3D model, however, the results do not support that conclusion. Moreover, the overall merit of the results seems not to be more proper for technical/methodological journals rather than for the IJMS aims and scope. Publication in IJMS requires sequential revision including experiments that will provide much more biological meaning, as well as clearly indicate future perspectives for its application for the study.

Reviewer 2 has an overall critical view on the study. However, we strongly disagree with his arguments. We see our printed constructs as models for a system with blood vessels. As we show in our study, we can connect the bioprinted model to a peristaltic pump and perfuse it over a long period. Furthermore, we carried out several experiments with biological (biomedical) meaning and tested an anticancer drug. We thus provide the proof-of-concept that the model is suitable for testing of new substances to treat cancer. We have discussed the topic of our study prior to submission with an editor of IJMS and identified a section (Molecular Oncology) and a special issue (Novel Molecular Pathways in Oncology) to which it fits well.

Even if the Reviewer was not convinced by the first draft, with the helpful suggestions of the other reviewers, we feel the reviewer’s reservations have largely been addressed, at least to the degree that the reviewer spelled them out. We therefore hope that the editor agrees that this study is of interest to the readership of the journal.

Furthermore, we are thankful to the other reviewers, who consider our study “an interesting approach … which could reduce the dependency on animal models” (reviewer 1). Furthermore, they state that the study is “a good concept that can benefit the researchers working in the area of cancer research and 3D cell culture” (reviewer 3) and “considered the data good, the experimental design correct, and the results coherent, and the conclusions are in line with the results obtained” (reviewer 4). The latter reviewer also considers the comparison between the 3D and 2D “very good”, the chemotherapeutic testing as “a good approach” and the perfusion “extremely relevant”.

Reviewer 3 Report

The manuscript entitled, “Generation of a perfusable 3D lung cancer model by Digital Light Processing is a good concept that can benefit the researchers working in the area of cancer research and 3D cell culture. However, the manuscript needs some work before considering for publication in International Journal of Molecular Sciences. Therefore, major revision is recommended concerning the science, quality of draft, and referencing. The suggestions are as follows:

Abstract

The result in terms of numbers should be presented in the abstract.

Line 23: How much higher? Mention the number while stating the significance.

Line 26: Can be used? Does it need any further improvement? What are the shortcomings of the current model? Please shed light on these points briefly.

Introduction

Line 34: Please provide reference.

Line 50-52: Rephrase the sentence.

Line 56: Please connect the current and subsequent paragraph in a better way.

Line 60: Please add reference.

Results

Figure 1 and 3: All (1a-1c and 3a-3c) are very small for the reader to interpret and correlate with information provided in the figure legend.

Figure 4: Please use similar units in both the figures, either nM or µM.

Figure 5: I cannot see 5c in the legend.

Figure 6: Please increase the size of 6a and 6c. It is very difficult to interpret the images.

Discussion

Line 288: Delete ‘over’, as it appears twice in the sentence.

Current studies are not referred as well as not included in the discussion and results for comparison.

Please update the discussion with the contemporary models in detail.

Materials and methods

Line 362: How 3D model was evaluated using fluorescent microscope? Why wasn’t confocal imaging preferred to have a 3D model a.k.a. Z-stack to have a better idea of how cells are growing in the 3D model.

Line 455: Delete ‘space’ after 10 µM.

Conclusion

Line 481: what is meant by other groups, provide more information.

The conclusion ends abruptly, please provide a future direct and prospects of the current study and concept, respectively.

Reference

The references are not up-to-date.

Reference 1 and 2 are same.

I would like to re-review the manuscript once the it is improved and comments are addressed.

Author Response

Reviewer 3

The manuscript entitled, “Generation of a perfusable 3D lung cancer model by Digital Light Processing” is a good concept that can benefit the researchers working in the area of cancer research and 3D cell culture. However, the manuscript needs some work before considering for publication in International Journal of Molecular Sciences. Therefore, major revision is recommended concerning the science, quality of draft, and referencing. The suggestions are as follows:

Thank you for considering our study a good concept with benefit for the field. We have included the suggestions as follows:

Abstract

The result in terms of numbers should be presented in the abstract.

Line 23: How much higher? Mention the number while stating the significance.

We agree that more scientific content was necessary in the abstract and implemented this constructive suggestion. We added details about the differences of the IC50 values for 3D and 3D cultures (three orders of magnitude). We also state that perfusion enhances cell viability by 60% and that the fraction of cells positive for the apoptosis markers Caspse-3 and PARP-1 is four- and six-fold higher, respectively.

Line 26: Can be used? Does it need any further improvement? What are the shortcomings of the current model? Please shed light on these points briefly.

This is another good suggestion to improve the abstract. The last two sentences of the abstract now read as follows:

Altogether, this easily reproducible cancer model can be used to for an initial testing of the cytotoxicity of new anticancer substances. For subsequent in-depth characterization of candidate drugs further improvements will be necessary, such as the generation of a multi-cell type lung cancer model and the lining of vascular structures with endothelial cells.

Introduction

Line 34: Please provide reference.

We provided a reference [1] for this statement, as requested.

Line 50-52: Rephrase the sentence.

We rephrased the sentence: In animal models, human tumors are embedded in a microenvironment composed of animal cells [14]. This chimeric constellation is unlikely to represent the physiology in a human patient.

Line 56: Please connect the current and subsequent paragraph in a better way.

Thank you. We completely re-phrased the start of the second paragraph to have a better connection: Although the reasons for terminating clinical trials are manifold, the high failure rate illustrates the need for better models for pre-clinical characterization of potential anti-cancer drugs.

A particularly promising strategy to produce models with higher relevance for human (patho-)physiology is the generation of organ models by 3D bioprinting which allows the arrangement of different types of (human) cells of an organ with high spatial resolution is a promising technology to produce organ models composed of human cells [15-17].

Line 60: Please add reference.

We found a review from 2023 which nicely explains DLP, as requested.

Results

Figure 1 and 3: All (1a-1c and 3a-3c) are very small for the reader to interpret and correlate with information provided in the figure legend.

We fully agree. We increased the size to the maximal width of the page. Some adjustments of the manuscript will be necessary for the final layout.

Figure 4: Please use similar units in both the figures, either nM or µM.

In 2D the drug is active in the low nanomolar range, while micromolar concentrations are needed for the 3D model. We tried to represent this with the same scale for both experiments but found that it is not possible. Thus, although we agree that identical axis are in general preferably, this is not possible here. To make the reader aware of this difference, we stressed the units in the revised figure legend.

Figure 5: I cannot see 5c in the legend.

Thank you for making us aware of this mistake which occurred after re-arranging the figure prior to submission. The figure parts d, e, and f should be c, d, and e. We corrected the legend.

Figure 6: Please increase the size of 6a and 6c. It is very difficult to interpret the images.

We re-arranged the figure so that the fluorescence microscopy images are now larger.

Discussion

Line 288: Delete ‘over’, as it appears twice in the sentence.

Thank you – deleted.

Current studies are not referred as well as not included in the discussion and results for comparison.

Please update the discussion with the contemporary models in detail.

We agree that our discussion needed improvements. As can be seen in the re-submitted manuscript, large parts of the discussion have been revised. We have added substantial information on the current state of the bioprinting approaches, contemporary models, strengths and challenges of bioprinted tumor models and tasks for the field in the near future. Numerous new recently published papers were discussed and added to the reference list (see also below)

Materials and methods

Line 362: How 3D model was evaluated using fluorescent microscope? Why wasn’t confocal imaging preferred to have a 3D model a.k.a. Z-stack to have a better idea of how cells are growing in the 3D model.

We have done many z-stacks on many of our bioprinted models. In addition, we took confocal images with the sophisticated microscopes of our collaborators. However, we did not gain additional insight into the live/dead stains. Due to the optical properties of the bioinks, it is very complicated to image deeper layers of the models. We have therefore retained the images taken with the inverted fluorescence microscope and rephrased the sentence in the Materials and Methods section to make this step clearer:

Analysis of the 2D cells or 3D models using an inverted fluorescence microscope (Observer Z1, (Zeiss, Jena, Germany) was conducted after a 1 h incubation in phenol red-free RPMI medium containing 2 µM calcein-AM and 2 µM ethidium homodimer-1.

Line 455: Delete ‘space’ after 10 µM.

Thank you – deleted.

Conclusion

Line 481: what is meant by other groups, provide more information.

We have now explained what we were aiming at: Importantly, the model can easily be adapted by other groups working in the field of cancer research and 3D culture as we provide all information to reproduce the model with commercially available devices and materials.

The conclusion ends abruptly, please provide a future direct and prospects of the current study and concept, respectively.

Thank you. The end was, in fact abrupt. We added two sentences:

We therefore consider the model to be suitable for testing new substances with presumed anti-tumoral activity. Further improvements of the model, such as endothelial lining, will be included in the next steps to increase its physiological relevance further.

Reference

The references are not up-to-date.

Thank you, this was a valuable advice. We updated the contents and reference list. We deleted numerous older references which could be replaced by more recent ones (former refs. No. 1, 3, 19, 39, 41, and 43). In total, 16 new references mostly from 2021-2023 were added. We agree that this was a necessary step to update the reference list and think that our manuscript is now up-to-date.

Reference 1 and 2 are same.

Actually, these were two different references from 2021 and 2022. In updating the references, we deleted the older one, which eliminates the risk of this sort of confusion.

Reviewer 3 made a number of suggestions, which we included in our manuscript. We think that this helped to improve the quality of our manuscript. Thank you.

Author Response

The paper describes the development of a 3D lung cancer model to test new drugs. This topic is very important and up to date, since currently it is well known that researchers need more in vivo like models to test new compounds. Thus, the most promising drugs are well tested, avoiding wasting time and money with substances that are not that efficacious. The paper is well written and the comparison between the 3D and 2D is very good. The use of a chemotherapeutic agent as a proof of concept for the value of the model was also a good approach. Additionally, authors added perfusion to the model, which demonstrated to be extremely relevant.Overall, I considered the data good, the experimental design correct, and the results coherent, and the conclusions are in line with the results obtained.

Thank you for the very positive evaluation of our manuscript. We included your minor comments as follows:

Minor comments:

Line 42 – Resistance to immunotherapy? Can you elaborate a bit on the topic?

We elaborated on this highly relevant topic and included a reference for further details:

But despite promising initial clinical outcomes in many cases, patients may eventually develop resistance to immunotherapy. The mechanism resulting in resistance to immunotherapy have not been fully understood, but include lack of tumor immunogenicity, deficiency in antigen presentation and aberrations in signaling pathways as well as tumor extrinsic mechanisms involved in the T cell activities.

Wang, F.; Wang, S.; Zhou, Q., The Resistance Mechanisms of Lung Cancer Immunotherapy. Front Oncol 2020, 10, 568059.

Line 79 – Why did you select H358 cell line, instead, for instance A549, which is one of the most used NSCLC secondary cell line?

We have used A549 cells in previous studies of our group (Berg et al., 2018 and 2021). Both cell lines harbor different KRAS mutations (G12C in the case of H358 and G12S in the case of A549). As a consequence, both cell lines have different sensitivity towards inhibitors. For example, only the H358 cell line, but not the A549 cells, were found to be sensitive to a MEK inhibitor [doi: 10.1002/mc.20607]. We wanted to establish a model to study such substances in future experiments and therefore used H358 cells. We included a new sentence and the appropriate reference.

Line 81 – you already described non-small cell lung cancer (NSCLC)

Thank you, we now use the abbreviation instead of writing it out.

Line 86 – Rephrase it!

The sentence was rephrased into: Dead cells made up a large fraction of the total number at the highest concentration used (64 nM), as indicated by the strong red fluorescence.

Line 134 – Why did you change the concentration of gentamicine from nM to uM? The effect on 3D model, decrease that much the efficacy of the drug?

Yes, this was one of the most surprising findings of the study, that the IC50 values were three orders of magnitude higher in the 3D model compared to the 2D culture. The most likely explanation is that the substances cannot easily penetrate the 3D constructs. We thus had to increase the concentration for studies in 3D culture. As this point was also raised by reviewer 1 (point 3), we elaborate on this topic in more detail in the discussion section.

Round 2

Reviewer 1 Report

I am happy with the correction. 

Reviewer 3 Report

The authors have amended the revised version based on the suggestions. The manuscript can be accepted provided comments from the editor and other reviewers are duly addressed.